# Effectiveness of nirmatrelvir/ritonavir and molnupiravir in reducing the risk of short-term and long-term cardiovascular complications of COVID-19: a target trial emulation study

While treatment with nirmatrelvir/ritonavir or molnupiravir is effective in lowering the rate of severe COVID-19, the effectiveness of these antivirals in reducing the risk of cardiovascular outcomes, especially among the hospitalized population, remains largely unknown. In this study, we assessed the real-world effectiveness of nirmatrelvir/ritonavir and molnupiravir on short- and long-term cardiovascular complications of COVID-19 using a target trial emulation design. Two target trials of COVID-19 antivirals were emulated by using a territory-wide, population-based, retrospective cohort of hospitalized patients in Hong Kong. Nine cardiovascular outcomes were evaluated in both short-term (day 0–21) and long-term (day 22–365) post-SARS-CoV-2 infection. Compared with the control group, the use of nirmatrelvir/ritonavir was associated with a significantly lower one-year risk of cardiovascular mortality, composite cardiovascular complications, major adverse cardiac events, cerebrovascular disorders, dysrhythmia, ischemic heart disease, and other cardiac disorders following infection. Molnupiravir use was associated with a short-term risk reduction in cardiovascular complications, but only a marginal risk reduction in long-term cardiovascular mortality among other complications. This study demonstrated the effectiveness of nirmatrelvir/ritonavir in reducing the risks of short- and long-term cardiovascular complications following a SARS-CoV-2 infection among the hospitalized population. Our findings suggested health-related benefits of prescribing nirmatrelvir/ritonavir over molnupiravir against severe cardiovascular post-acute sequelae of COVID-19 in the long term.

Nirmatrelvir/ritonavir and molnupiravir have been authorized for COVID-19 treatment since 2021. Randomized controlled trials and observational studies have shown their short-term benefits in reducing the rate of mortality and hospitalization for high-risk outpatients,[1–3] yet the effectiveness in other conditions remains unclear due to mixed findings.[4,5]

It is well-documented that SARS-CoV-2 infection was associated with cardiovascular events occurring during the acute phase of

✉e-mail: zhaoshi.cmsa@gmail.com; yeoh_ek@cuhk.edu.hk; marc@cuhk.edu.hk

infection and the post-acute phase of COVID-19,[6,7] resulting in a decline in quality of life. According to investigations using the Veterans cohort of the United States,[8,9] long-term cardiovascular (e.g., ischemic and non-ischemic heart disease, myocarditis, and heart failure) and cerebrovascular complications (e.g., ischemic and hemorrhagic stroke, cerebral venous thrombosis, and transient ischemic attacks) were more likely to occur among patients recovered from COVID-19, compared to historical controls. While studies demonstrated that the antivirals were associated with a lower risk of cardiovascular diseases in non-hospitalized COVID-19 patients with autoimmune rheumatic diseases[10] and kidney diseases,[11] none have examined the effectiveness of the two antivirals in the hospitalized population, who presented more severe COVID-19 in the general population.

Several randomized controlled trials have been initiated to evaluate nirmatrelvir/ritonavir as a potential treatment for post-acute sequelae of COVID-19.[12–15] The RECOVER-Vital,[12] PAX LC trial,[13] and STOP-PASC trial[14] focused on patients with long COVID, while the CanTreatCOVID involved patients with acute infection.[15] However, these trials only focused on non-specific conditions. To date, there is no randomized controlled trial specifically investigating the efficacy of antiviral agents on cardiovascular complications in COVID-19 hospitalizations. Here, we used a target trial emulation study design to evaluate the effectiveness of nirmatrelvir/ritonavir and molnupiravir on acute and post-acute COVID-19 cardiovascular complications among the hospitalized population.

## Results

During the study period from March 11, 2022, to October 10, 2023, we identified a total of 71,258 hospitalized individuals who were first-time infected with SARS-CoV-2 (Fig. 1). After screening, 14,842 patients who received nirmatrelvir/ritonavir and 19,660 controls were eligible for inclusion in the nirmatrelvir/ritonavir trial; and 10,053 patients who received molnupiravir and 22,163 controls eligible for inclusion in the molnupiravir trial.

The baseline characteristics of patients included in the two trials are shown in Table 1. In the nirmatrelvir/ritonavir trial, nirmatrelvir/ritonavir recipients and controls had a median age of 75 (IQR: 66–84) years and 72 (55–85) years, with 7344 (49.5%) and 10,124 (51.5%) female

patients, respectively. For the molnupiravir trial, molnupiravir recipients and controls had a median age of 78 (IQR: 67–87) years and 73 (57–85) years, with 4894 (48.7%) and 11,289 (50.9%) female patients, respectively. After applying inverse probability of censoring weighting (IPCW), most of the baseline covariates were well balanced for the two trials, with an absolute standardized mean difference (SMD) less than 0.1 (Supplementary Fig. 1 and Supplementary Fig. 2). Covariates with an absolute SMD ≥0.1 were included in the corresponding models for doubly robust adjustment.

### Short-term outcomes

**Effect of nirmatrelvir/ritonavir.** The median follow-up period for patients in the nirmatrelvir/ritonavir trial was 21 days for short-term outcomes. The 21-day adjusted cumulative incidence of cardiovascular death, major adverse cardiovascular events (MACE), and composite cardiovascular complications is shown in Fig. 2. There were reductions in the risk of cardiovascular mortality (adjusted risk difference (RD) (95% confidence interval (CI)) = − 0.97% (−1.24 to −0.71); adjusted hazard ratio (HR) (95% CI) = 0.45 (0.28 to 0.72)), composite cardiovascular complications (adjusted RD = −3.02% (−3.64 to −2.41); adjusted HR = 0.55 (0.46 to 0.65)), MACE (adjusted RD = −2.13% (−2.58 to −1.67); adjusted HR = 0.47 (0.39 to 0.57)), cerebrovascular disorders (adjusted RD = −0.91% (−1.26 to −0.57); adjusted HR = 0.43 (0.31 to 0.60)), dysrhythmia (adjusted RD = −0.67% (−1.03 to −0.31); adjusted HR = 0.68 (0.56 to 0.83)), ischemic heart disease (adjusted RD = −1.14% (−1.45 to −0.83); adjusted HR = 0.38 (0.28 to 0.53)), and other cardiac disorders (adjusted RD = −0.87% (−1.12 to −0.63); adjusted HR = 0.51 (0.32 to 0.81)). The 21-day risk of inflammatory heart disease and thrombotic disorders was similar between arms (Fig. 3).

**Effect of molnupiravir.** The median follow-up period for patients in the molnupiravir trial was 21 days for short-term outcomes. Molnupiravir was associated with a lower risk of cardiovascular mortality (adjusted RD = −1.03% (−1.37 to −0.69); adjusted HR = 0.66 (0.53 to 0.82)), composite cardiovascular complications (adjusted RD = −1.36% (−2.03 to −0.70); adjusted HR = 0.80 (0.72 to 0.90)), MACE (adjusted RD = −1.02% (−1.52 to −0.53); adjusted HR = 0.75 (0.65 to 0.87)), cerebrovascular disorders (adjusted RD = −0.74% (−1.06 to −0.43); adjusted

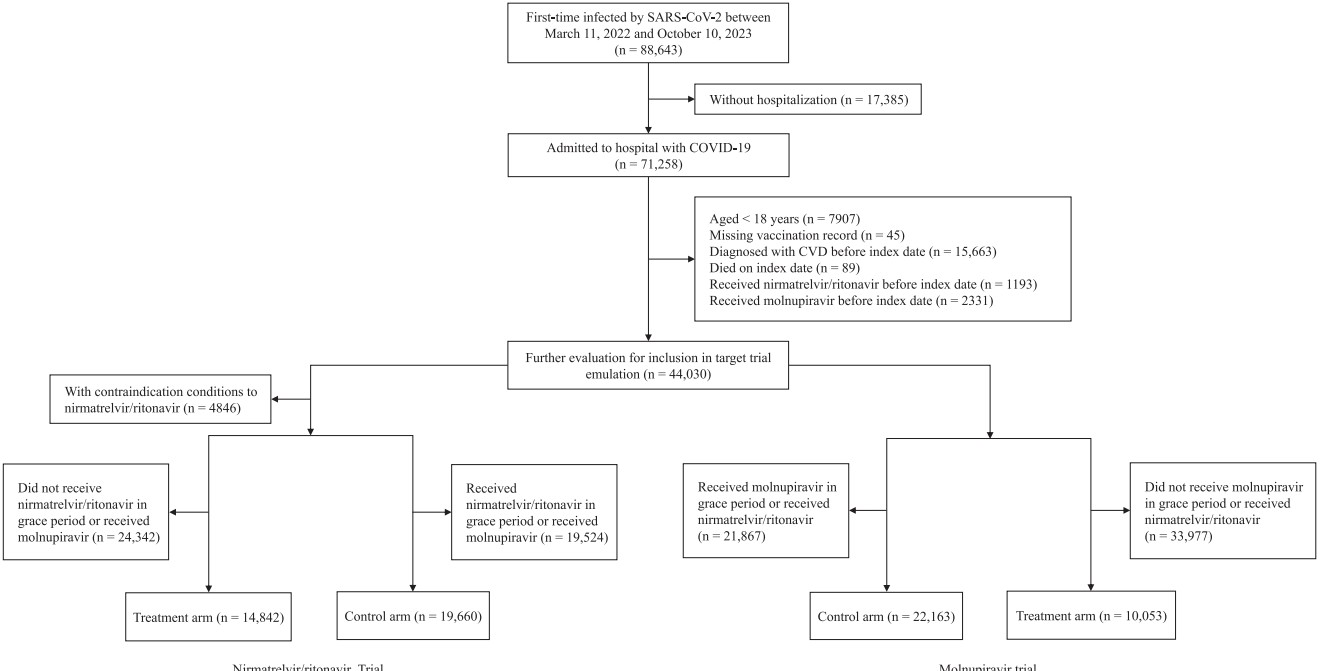

**Fig. 1 | Flowchart of patient inclusion and exclusion.** CVD cardiovascular diseases.

**Table 1 | Baseline characteristics of study participants in two emulated target trials**

| Characteristics | Nirmatrelvir/ritonavir trial | | Molnupiravir trial | |
|---|---|---|---|---|
| | Treatment arm (n = 14842) | Control arm (n = 19660) | Treatment arm (n = 10053) | Control arm (n = 22163) |
| Age, years | 75 [66–84] | 72 [55–85] | 78 [67–87] | 73 [57–85] |
| Sex | | | | |
| Female | 7344 (49.5) | 10,124 (51.5) | 4894 (48.7) | 11,289 (50.9) |
| Male | 7498 (50.5) | 9536 (48.5) | 5159 (51.3) | 10,874 (49.1) |
| Vaccination status | | | | |
| Unvaccinated | 2183 (14.7) | 5427 (27.6) | 2107 (21.0) | 6445 (29.1) |
| 1–2 doses | 2713 (18.3) | 6157 (31.3) | 2228 (22.2) | 6953 (31.4) |
| ≥3 doses | 9946 (67.0) | 8076 (41.1) | 5718 (56.9) | 8765 (39.5) |
| Charlson Comorbidity Index | 0 [0–1] | 0 [0–1] | 0 [0–1] | 0 [0–1] |
| Pre-existing conditions | | | | |
| Chronic pulmonary disease | 624 (4.2) | 1200 (6.1) | 493 (4.9) | 1335 (6.0) |
| Dementia | 143 (1.0) | 271 (1.4) | 217 (2.2) | 313 (1.4) |
| Hypertension | 6377 (43.0) | 6998 (35.6) | 4469 (44.5) | 8107 (36.6) |
| Rheumatic disease | 128 (0.9) | 179 (0.9) | 117 (1.2) | 199 (0.9) |
| Renal disease | 104 (0.7) | 222 (1.1) | 386 (3.8) | 361 (1.6) |
| Diabetes | 2786 (18.8) | 2820 (14.3) | 1804 (17.9) | 3260 (14.7) |
| Liver disease | 227 (1.5) | 401 (2.0) | 355 (3.5) | 610 (2.8) |
| Peptic ulcer disease | 34 (0.2) | 52 (0.3) | 20 (0.2) | 56 (0.3) |
| Cancer | 220 (1.5) | 300 (1.5) | 139 (1.4) | 339 (1.5) |
| Hemiplegia or paraplegia | 1 (<0.1) | 2 (<0.1) | 2 (<0.1) | 3 (<0.1) |
| HIV infection | 4 (<0.1) | 9 (<0.1) | 10 (0.1) | 12 (0.1) |
| Immunocompromised status | 899 (6.1) | 1390 (7.1) | 784 (7.8) | 1611 (7.3) |
| Concomitant pharmacological treatments | | | | |
| Dexamethasone | 1826 (12.3) | 6383 (32.5) | 2196 (21.8) | 7553 (34.1) |
| Methylprednisolone | 8 (0.1) | 36 (0.2) | 5 (<0.1) | 42 (0.2) |
| Prednisolone | 596 (4.0) | 1284 (6.5) | 619 (6.2) | 1473 (6.6) |
| Interferon | 13 (0.1) | 105 (0.5) | 34 (0.3) | 133 (0.6) |
| Baricitinib | 193 (1.3) | 504 (2.6) | 132 (1.3) | 615 (2.8) |
| Tocilizumab | 56 (0.4) | 289 (1.5) | 52 (0.5) | 350 (1.6) |
| Remdesivir | 1223 (8.2) | 4456 (22.7) | 1335 (13.3) | 5095 (23.0) |
| Concomitant non-pharmacological treatments | | | | |
| Intensive care unit admission | 206 (1.4) | 808 (4.1) | 163 (1.6) | 1030 (4.6) |
| Use of ventilatory support | 156 (1.1) | 719 (3.7) | 173 (1.7) | 936 (4.2) |

Data are presented in n (%) or median [IQR].

HR = 0.57 (0.44 to 0.76)), other cardiac disorders (adjusted RD = −0.66% (−0.99 to −0.34); adjusted HR = 0.73 (0.58 to 0.92)), and thrombotic disorders (adjusted RD = −0.16% (−0.29 to −0.02); adjusted HR = 0.54 (0.31 to 0.96)). The risk of dysrhythmia, ischemic heart disease, and inflammatory heart disease was similar between arms (Figs. 2 and 3).

## Long-term outcomes
**Effect of nirmatrelvir/ritonavir.** The median follow-up period for patients in the nirmatrelvir/ritonavir trial was 365 days for long-term outcomes. In general, the 22- to 365-day risk of cardiovascular outcomes was lower in the nirmatrelvir/ritonavir arm than in the control arm. Compared with control arm, the use of nirmatrelvir/ritonavir was associated with a significantly lower risk of cardiovascular mortality (adjusted RD = −0.81% (−1.19 to −0.43); adjusted HR = 0.61 (0.47 to 0.79)), composite cardiovascular complications (adjusted RD = −1.50% (−2.15 to −0.85); adjusted HR = 0.73 (0.63 to 0.86)), MACE (adjusted RD = −1.23% (−1.80 to −0.67); adjusted HR = 0.70 (0.56 to 0.87)), cerebrovascular disorders (adjusted RD = −0.53% (−0.89 to −0.16)); adjusted HR = 0.66 (0.51 to 0.86)), dysrhythmia (adjusted RD = −0.73%

(−1.10 to −0.35); adjusted HR = 0.63 (0.52 to 0.77)), ischemic heart disease (adjusted RD = −0.21% (−0.49 to 0.08); adjusted HR = 0.75 (0.60 to 0.95)), and other cardiac disorders (adjusted RD = −0.84% (−1.28 to −0.39); adjusted HR = 0.70 (0.51 to 0.95)). The statistical significance of inflammatory heart disease and thrombotic disorders reached the marginal level of 0.05 (Figs. 4 and 5).

**Effect of molnupiravir.** The median follow-up period for patients in the molnupiravir trial was 365 days for long-term outcomes. Compared to nirmatrelvir/ritonavir, the benefits of molnupiravir on cardiovascular outcomes were less apparent (Figs. 4 and 5). Among all the cardiovascular outcomes, molnupiravir was only marginally significantly associated with a decreased risk of cardiovascular mortality (adjusted RD = −0.37% (−0.78 to 0.03); adjusted HR = 0.81 (0.68 to 0.96)), compared to the control group (Fig. 5).

## Subgroup and sensitivity analyses
The effect sizes of nirmatrelvir/ritonavir and molnupiravir generally remained consistent with those observed in the primary analysis for patients aged over 65 years (Supplementary Fig. 3). However, for

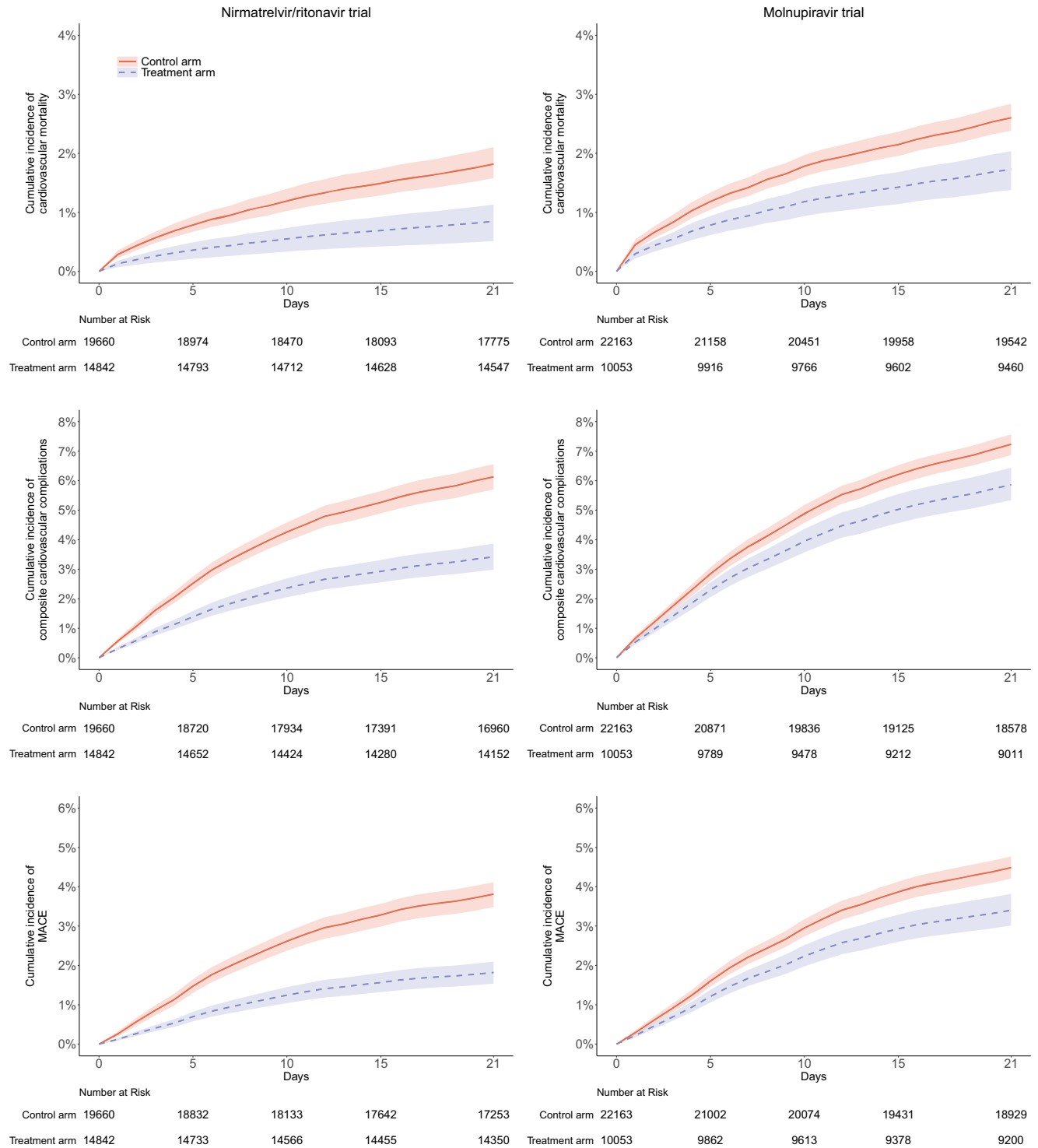

**Fig. 2 | Adjusted cumulative 21-day incidence curve of cardiovascular complications between treatment arm (blue dashed line) and control arm (red solid line) in target trials of patients hospitalized for COVID-19.** The lines represent the cumulative incidence adjusting for baseline covariates. Shaded regions indicate 95% confidence bands constructed using 1000 bootstrap resamples. Tables represent numbers at risk during the follow-up period. MACE major adverse cardiovascular events.

patients aged ≤ 65 years, the effect estimates differed from the primary analysis, albeit with wide confidence intervals (Supplementary Fig. 4). Similar results were found for male and female subgroups in two trials compared with results from main analyses (Supplementary Fig. 5 and Supplementary Fig. 6). While the results were generally in line with main findings in vaccinated patients, receiving nirmatrelvir/ritonavir was only associated with a lower long-term risk for composite

cardiovascular complications, MACE, and dysrhythmia in unvaccinated subgroup (Supplementary Fig. 7 and Supplementary Fig. 8).

Sensitivity analyses of varying the assessment period of cardiovascular outcomes (Supplementary Fig. 9), adjusting the historical period of the outcome events (Supplementary Fig. 10), and using the Fine-Gray model (Supplementary Fig. 11) yielded similar results as those found in the primary analyses.

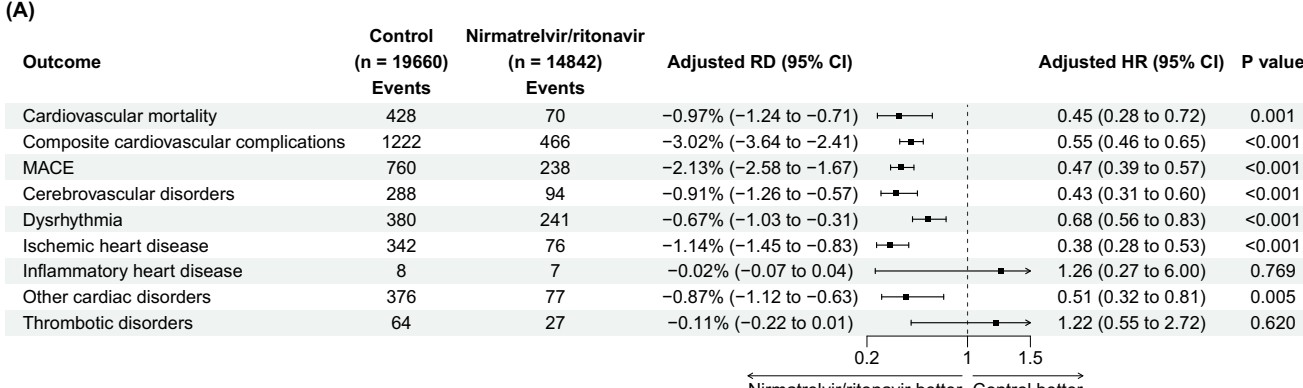

**(A)**

| Outcome | Control (n = 19660) Events | Nirmatrelvir/ritonavir (n = 14842) Events | Adjusted RD (95% CI) | Adjusted HR (95% CI) | P value |
|---|---|---|---|---|---|
| Cardiovascular mortality | 428 | 70 | −0.97% (−1.24 to −0.71) | 0.45 (0.28 to 0.72) | 0.001 |
| Composite cardiovascular complications | 1222 | 466 | −3.02% (−3.64 to −2.41) | 0.55 (0.46 to 0.65) | <0.001 |
| MACE | 760 | 238 | −2.13% (−2.58 to −1.67) | 0.47 (0.39 to 0.57) | <0.001 |
| Cerebrovascular disorders | 288 | 94 | −0.91% (−1.26 to −0.57) | 0.43 (0.31 to 0.60) | <0.001 |
| Dysrhythmia | 380 | 241 | −0.67% (−1.03 to −0.31) | 0.68 (0.56 to 0.83) | <0.001 |
| Ischemic heart disease | 342 | 76 | −1.14% (−1.45 to −0.83) | 0.38 (0.28 to 0.53) | <0.001 |
| Inflammatory heart disease | 8 | 7 | −0.02% (−0.07 to 0.04) | 1.26 (0.27 to 6.00) | 0.769 |
| Other cardiac disorders | 376 | 77 | −0.87% (−1.12 to −0.63) | 0.51 (0.32 to 0.81) | 0.005 |
| Thrombotic disorders | 64 | 27 | −0.11% (−0.22 to 0.01) | 1.22 (0.55 to 2.72) | 0.620 |

0.2   1   1.5
Nirmatrelvir/ritonavir better   Control better

**(B)**

| Outcome | Control (n = 22163) Events | Molnupiravir (n = 10053) Events | Adjusted RD (95% CI) | Adjusted HR (95% CI) | P value |
|---|---|---|---|---|---|
| Cardiovascular mortality | 583 | 143 | −1.03% (−1.37 to −0.69) | 0.66 (0.53 to 0.82) | <0.001 |
| Composite cardiovascular complications | 1520 | 586 | −1.36% (−2.03 to −0.70) | 0.80 (0.72 to 0.90) | <0.001 |
| MACE | 939 | 336 | −1.02% (−1.52 to −0.53) | 0.75 (0.65 to 0.87) | <0.001 |
| Cerebrovascular disorders | 336 | 93 | −0.74% (−1.06 to −0.43) | 0.57 (0.44 to 0.76) | <0.001 |
| Dysrhythmia | 493 | 243 | −0.14% (−0.55 to 0.26) | 0.91 (0.77 to 1.08) | 0.285 |
| Ischemic heart disease | 429 | 201 | −0.04% (−0.40 to 0.32) | 0.95 (0.78 to 1.15) | 0.603 |
| Inflammatory heart disease | 12 | 4 | −0.04% (−0.10 to 0.03) | 0.99 (0.26 to 3.72) | 0.982 |
| Other cardiac disorders | 489 | 134 | −0.66% (−0.99 to −0.34) | 0.73 (0.58 to 0.92) | 0.007 |
| Thrombotic disorders | 78 | 19 | −0.16% (−0.29 to −0.02) | 0.54 (0.31 to 0.96) | 0.036 |

0.2   1   1.5
Molnupiravir better   Control better

**Fig. 3 | Risk of cardiovascular complications at day 21 in target trials of patients hospitalized for COVID-19. A** Target trial of nirmatrelvir/ritonavir versus no treatment. **B** Target trial of molnupiravir versus no treatment. Adjusted HRs (square dots) and 95% CIs (error bars) are presented in (**A**) and (**B**). The dashed vertical line in (**A**) and (**B**) represents the HR of 1.00. Statistical analysis with two-sided Wald test in (**A**) and (**B**). MACE major adverse cardiovascular events. RD risk difference. HR hazard ratio. CI confidence interval.

## Discussion

Cardiovascular complications are common conditions following SARS-CoV-2 infection, resulting in a substantial health burden in healthcare systems.[8,9] Nirmatrelvir/ritonavir and molnupiravir have been shown to effectively reduce the acute severity of COVID-19 among various populations.[10] Considering the acute severity of COVID-19 (e.g., required hospitalization care[16]) is a general risk factor for COVID-19 prognosis, our study hypothesized that antivirals, including nirmatrelvir/ritonavir and molnupiravir, are effective in lowering the risk of cardiovascular complications of COVID-19 among the hospitalized population. According to our target trial emulation, hospitalized patients with COVID-19 who initiated nirmatrelvir/ritonavir had lower risks of both short- and long-term cardiovascular complications, compared to non-initiators. In contrast, while a similar short-term risk reduction in cardiovascular complications was observed in molnupiravir-treated patients, our study showed the risk reduction associated with molnupiravir was limited to cardiovascular mortality among all the long-term study outcomes.

Our study primarily demonstrated that nirmatrelvir/ritonavir is effective in lowering the risk of cardiovascular complications one year following SARS-CoV-2 infection among the hospitalized population, which is consistent with two similar studies involving other patient populations.[10,11] Wang et al. showed that nirmatrelvir/ritonavir significantly reduced the risks of cerebrovascular complications, arrhythmia, ischemic heart disease, other cardiac disorders, and MACE among patients with autoimmune rheumatic diseases,[10] whereas Strohbehn et al. indicated that nirmatrelvir/ritonavir-treated patients had a lower risk of hospitalization for MACE among those with advanced chronic kidney disease and kidney failure.[11] Despite involving different study populations, these works included patients who are generally more vulnerable to the acute severity of COVID-19 than outpatients. Given that COVID-19 severity is a risk factor for long-term cardiovascular complications,[17] the mitigation effect of nirmatrelvir/ritonavir on acute illness and viral replication may thus explain the observed risk reduction of cardiovascular complications during both the acute and post-acute phases of COVID-19 in these populations.

Among the cardiovascular complications studied, we demonstrated that the prescription of nirmatrelvir/ritonavir was associated with reduced risks of MACE, cerebrovascular disorders, dysrhythmia, ischemic heart disease, and other cardiac disorders. The benefits of nirmatrelvir/ritonavir for these complications are also identified in Wang et al.,[10] with similar effect sizes for long-term cerebrovascular disorders (adjusted HR = 0.66), even though the incidences of post-COVID cardiovascular outcomes are generally higher in their population with autoimmune rheumatic diseases. Regarding MACE, our estimate of short-term adjusted HR (0.47) is also consistent with that of Strohbehn et al.[11] (0.49), which was assessed among patients with kidney diseases. Our estimate of adjusted HR for long-term MACE (0.70) is similar to that of Liu et al.[17] (0.65), whose study included only a non-hospitalized population.

Previous randomized controlled trials[1,18,19] and observational studies[17,20–22] have demonstrated a smaller effect size for molnupiravir than for nirmatrelvir/ritonavir against conditions in both the acute and post-acute phases of COVID-19. In our study, although the short-term benefit of molnupiravir was found for cardiovascular complications, such an effect did not persist for most long-term cardiovascular outcomes. This finding was in line with Bajema et al.[21] and Wei et al.,[22] suggesting that the effect of molnupiravir on post-acute COVID-19

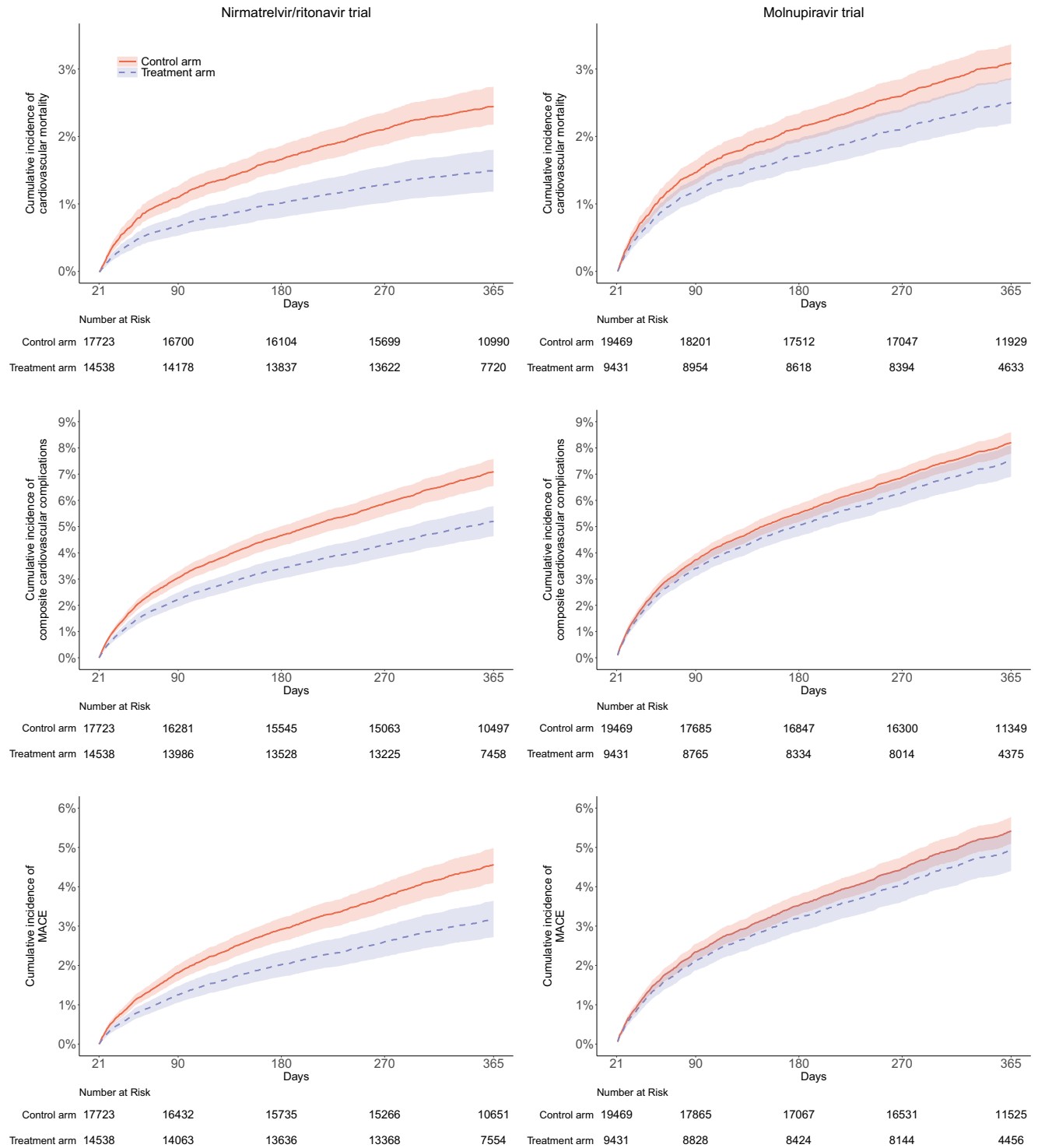

**Fig. 4 | Adjusted cumulative 22- to 365-day incidence curve of cardiovascular complications between treatment arm (blue dashed line) and control arm (red solid line) in target trials.** The lines represent the cumulative incidence adjusting for baseline covariates. Shaded regions indicate 95% confidence bands constructed using 1000 bootstrap resamples. Tables represent numbers at risk during the follow-up period. MACE major adverse cardiovascular events.

outcomes may be more limited compared to nirmatrelvir/ritonavir. Supported by Liu et al.,[17] the inferiority of molnupiravir compared to nirmatrelvir/ritonavir for long-term cardiovascular risk was also shown in non-hospitalized patients with COVID-19, although those prescribed molnupiravir still had a lower risk of one-year MACE and heart failure. Xie et al.[23] showed that the prescription of molnupiravir was associated with a reduced risk of dysrhythmia, but not of ischemic heart disease. Nevertheless, we acknowledge that molnupiravir may be provided for patients with severe renal and hepatic impairment, conditions for which nirmatrelvir/ritonavir is not indicated. The residual confounding of these critical conditions may diminish the effect size of molnupiravir in our investigation.

The major strength of our investigation is the use of a large population cohort, primarily infected with the Omicron variant of SARS-CoV-2, which reduces the impact of variant heterogeneity[24] and reinfections exhibited in other large cohorts of long COVID. Under

**(A)**

| Outcome | Control (n = 17723) Events | Nirmatrelvir/ritonavir (n = 14538) Events | Adjusted RD (95% CI) | | Adjusted HR (95% CI) | P value |
|---|---|---|---|---|---|---|
| Cardiovascular mortality | 430 | 185 | −0.81% (−1.19 to −0.43) | | 0.61 (0.47 to 0.79) | <0.001 |
| Composite cardiovascular complications | 1174 | 693 | −1.50% (−2.15 to −0.85) | | 0.73 (0.63 to 0.86) | <0.001 |
| MACE | 743 | 429 | −1.23% (−1.80 to −0.67) | | 0.70 (0.56 to 0.87) | 0.001 |
| Cerebrovascular disorders | 264 | 179 | −0.53% (−0.89 to −0.16) | | 0.66 (0.51 to 0.86) | 0.002 |
| Dysrhythmia | 386 | 235 | −0.73% (−1.10 to −0.35) | | 0.63 (0.52 to 0.77) | <0.001 |
| Ischemic heart disease | 279 | 172 | −0.21% (−0.49 to 0.08) | | 0.75 (0.60 to 0.95) | 0.017 |
| Inflammatory heart disease | 23 | 10 | −0.06% (−0.12 to 0.00) | | 0.47 (0.21 to 1.05) | 0.067 |
| Other cardiac disorders | 499 | 243 | −0.84% (−1.28 to −0.39) | | 0.70 (0.51 to 0.95) | 0.024 |
| Thrombotic disorders | 112 | 59 | −0.10% (−0.28 to 0.08) | | 0.70 (0.46 to 1.04) | 0.078 |

0.2  1  1.5

Nirmatrelvir/ritonavir better  Control better

**(B)**

| Outcome | Control (n = 19469) Events | Molnupiravir (n = 9431) Events | Adjusted RD (95% CI) | | Adjusted HR (95% CI) | P value |
|---|---|---|---|---|---|---|
| Cardiovascular mortality | 537 | 234 | −0.37% (−0.78 to 0.03) | | 0.81 (0.68 to 0.96) | 0.017 |
| Composite cardiovascular complications | 1416 | 692 | −0.29% (−1.00 to 0.43) | | 0.91 (0.82 to 1.01) | 0.090 |
| MACE | 913 | 464 | −0.24% (−0.85 to 0.38) | | 0.91 (0.79 to 1.04) | 0.160 |
| Cerebrovascular disorders | 320 | 162 | −0.18% (−0.55 to 0.20) | | 0.87 (0.70 to 1.09) | 0.221 |
| Dysrhythmia | 470 | 261 | −0.10% (−0.53 to 0.32) | | 0.92 (0.77 to 1.09) | 0.335 |
| Ischemic heart disease | 341 | 177 | 0.03% (−0.31 to 0.38) | | 0.95 (0.77 to 1.17) | 0.642 |
| Inflammatory heart disease | 27 | 10 | 0.01% (−0.08 to 0.10) | | 1.04 (0.46 to 2.36) | 0.927 |
| Other cardiac disorders | 624 | 306 | −0.12% (−0.62 to 0.37) | | 0.91 (0.78 to 1.08) | 0.285 |
| Thrombotic disorders | 133 | 53 | −0.10% (−0.30 to 0.10) | | 0.79 (0.56 to 1.13) | 0.202 |

0.2  1  1.5

Molnupiravir better  Control better

**Fig. 5 | Risk of cardiovascular complications from 22 to 365 days target trials of patients hospitalized for COVID-19. A** Target trial of nirmatrelvir/ritonavir versus no treatment. **B** Target trial of molnupiravir versus no treatment. Adjusted HRs (square dots) and 95% CIs (error bars) are presented in (**A**) and (**B**). The dashed vertical line in (**A**) and (**B**) represents the HR of 1.00. Statistical analysis with two-sided Wald test in (**A**) and (**B**). MACE major adverse cardiovascular events. RD risk difference. HR hazard ratio. CI confidence interval.

strict public health measures since the pandemic began in 2020, Hong Kong is a setting where over 99% of SARS-CoV-2 infections occurred during the Omicron outbreak.[25,26] Considering the ongoing trend of stable or reduced acute severity as well as the duration of acute phase across SARS-CoV-2's genetic variants and sub-lineages,[24] our findings inferred from an Omicron-dominated population are likely to hold for newly emerging variants. In addition, our study utilized all hospitalization records from the public sector, which accounts for approximately 90% of hospital admissions in Hong Kong. Given the close clinical monitoring during inpatient stays, the validity of the study data, particularly the antiviral prescription records, was fully captured.

This study presents several limitations. Firstly, the study population was hospitalized patients with COVID-19. Therefore, the results cannot be generalized to patients with mild COVID-19 (e.g., outpatients). As the eligibility criteria for antivirals are patients with moderate to severe symptoms, emulating a target trial among non-hospitalized patients could be challenging, as the assumption of exchangeability is difficult to achieve due to confounding by indication. Secondly, the low event counts of inflammatory heart disease and thrombotic disorders in our investigation may introduce statistical issue regarding sparse data bias, and studies with larger event observations are warranted to provide higher statistical power and precision for plausible results. Due to this limitation, we did not investigate the sub-categories of the diseases, such as myocardial infarction of ischemic heart disease. Thirdly, despite using the target trial framework on real-world data, the lack of available information on some confounders could lead to residual confounding, affecting the implementation of this trial emulation. Fourthly, although the median age of our study participants was consistent with that of the general

population hospitalized for COVID-19 in Hong Kong (77 years (IQR: 66–87) obtained from the original database), future study conducted in a younger population is warranted to validate our findings. In addition, as the absence of cardiovascular diseases was one of the inclusion criteria, the prevalence of comorbidities of study participants would be lower than the general population, given that cardiovascular diseases were associated with conditions including chronic kidney disease, diabetes, and certain cancers. Lastly, our study did not examine other antivirals, such as remdesivir, due to the relatively low number of recipients.

In conclusion, our target trial emulation demonstrates the effectiveness of nirmatrelvir/ritonavir in reducing the risks of short- and long-term cardiovascular complications following a SARS-CoV-2 infection among the hospitalized population. Given the lack of protective effect of molnupiravir on most long-term cardiovascular outcomes, we suggest prescribing nirmatrelvir/ritonavir over molnupiravir to prevent severe cardiovascular post-acute sequelae of COVID-19.

## Methods
### Study design
We conducted a territory-wide, population-based, retrospective cohort study to emulate two target randomized controlled trials of COVID-19 antivirals in hospitalized patients with first positive SARS-CoV-2 RT-PCR (reverse transcription polymerase chain reaction) test results from March 11, 2022 to October 10, 2023 (21 days before the end of the data availability date, which is October 31, 2023). Traditional cohort study design using observational data may be subject to sample selection bias and immortal-time bias due to the misalignment of

eligibility criteria and treatment assignment, and thus resulting in deviations from the true causal effect. The adoption of target trial emulation methods can mitigate such biases while preserving the advantages of large-scale observational databases, and further strengthening the comparability of treatment and control groups.[27,28] The target trials involved nirmatrelvir/ritonavir initiators versus non-initiators (nirmatrelvir/ritonavir trial), and molnupiravir initiators versus non-initiators (molnupiravir trial). The specification of the target trials was detailed in Supplementary Table 1.

## Study setting and Data sources
Anonymized individual-level electronic health records from the Hospital Authority (HA) and the Department of Health (DH) in Hong Kong were used in this study. As a statutory body, the HA delivers public inpatient and outpatient care, serving over 7.3 million residents and handling more than 90% of all hospitalizations in the region. Patient's demographic characteristics, death registry, inpatient and outpatient records, laboratory test records, and medication prescription records were collected from the HA database. DH provided de-identified population-based vaccination records, which were matched to the HA database.

The study followed the TARGET (Transparent Reporting of Observational Studies Emulating a Target Trial)[29] and the STROBE (Strengthening the Reporting of Observational Studies in Epidemiology) reporting guidelines. Ethics approval was obtained from the Joint CUHK-NTEC Clinical Research Ethics Committee (No.: 2023.006). No protocol was registered for this study.

## Study participants
We included patients aged 18 years or older who were first-time infected with SARS-CoV-2 and confirmed by positive reverse transcription polymerase chain reaction (RT-PCR) testing results. SARS-CoV-2 Omicron variants were the dominant variants during the patients' enrollment period, from March, 2022 to October, 2023. Patients who were admitted 3 days before or after the positive RT-PCR date were considered hospitalizations with COVID-19, and thus were eligible for inclusion.[30–33] These inclusion criteria also took into account the possible delay between case confirmation and hospital admission during the growth phase of the epidemic. The index date was defined as the earliest calendar date when the subject was test-positive for SARS-CoV-2 infection. The indication of nirmatrelvir/ritonavir and molnupiravir was initiated within 5 days after the symptom onset (the index date was used as a proxy of the symptom onset date). Patients who were prescribed either of the antiviral drugs before the index date were excluded. Patients with the following contraindications to nirmatrelvir/ritonavir were additionally excluded from nirmatrelvir/ritonavir trial: (1) drug contraindications (e.g., amiodarone, lumacaftor–ivacaftor, rifampicin, apalutamide, phenobarbital, rifapentine, carbamazepine, phenytoin, St John's Wort [hypericum perforatum], ivosidenib, and primidone) within 90 days of the index date; (2) severe renal impairment (i.e., estimated glomerular filtration rate <30 mL/min per 1.73 m$^2$, dialysis, or renal transplantation); and (3) severe liver impairment (i.e., cirrhosis, hepatocellular carcinoma, or liver transplantation).[3,4,20]

## Outcomes
The cardiovascular outcomes included in our study were in line with previous studies on cardiovascular complications of COVID-19 and post-COVID conditions,[8,10,34] including cerebrovascular disorders (stroke and transient ischemic attack), dysrhythmia (atrial fibrillation, tachycardia, bradycardia, and ventricular arrhythmia), ischemic heart disease (acute coronary disease, ischemic cardiomyopathy, angina, and myocardial infarction), inflammatory heart disease (pericarditis, and myocarditis), thrombotic disorders (pulmonary embolism, deep vein thrombosis, and superficial vein thrombosis), other cardiac

disorders (heart failure, non-ischemic cardiomyopathy, cardiac arrest, and cardiogenic shock), and MACE (myocardial infarction, stroke, heart failure, ventricular arrhythmia, and cardiac arrest). We defined composite cardiovascular complication as the presence of any of the aforementioned cardiovascular outcomes of interest. In addition, we considered cardiovascular mortality as an outcome of severe cardiovascular complications, defined as a death within 30 days following the diagnosis of any aforementioned cardiovascular outcomes.

Cardiovascular outcomes were ascertained using the International Classification of Diseases, Ninth Revision, Clinical Modification (ICD-9-CM) code or the International Classification of Primary Care, Second Edition (ICPC-2) code, which were defined in Supplementary Table 2. Patients with these cardiovascular outcomes diagnosed three years before baseline were excluded. Patients with prior diagnoses of the outcome of interest were excluded from the analyses of the particular condition. To avoid misattributing acute outcomes to the post-acute phase, we assessed cardiovascular outcomes in the short-term (from day 0 to day 21) and the long-term (from day 22 to 365), with the long-term analysis including only patients who were alive at day 21.[5,21,35] Patients were followed from the baseline until the occurrence of outcome events, death, the end of the outcome assessment period, or the end of data availability, whichever came first.

## Covariates
We collected the baseline characteristics of eligible patients, including age, sex, Charlson Comorbidity Index (calculated based on diagnosis on index date), pre-existing conditions on index date (chronic pulmonary disease, dementia, hypertension, HIV infection, cancer, liver disease, renal disease, diabetes, peptic ulcer disease, hemiplegia or paraplegia, and rheumatic disease), immunocompromised status on index date, admission to intensive care unit and ventilation support (Supplementary Table 3) within three days following the index date, initiation of concomitant treatments (dexamethasone, methylprednisolone, prednisolone, interferon-beta-1b, baricitinib, tocilizumab, and remdesivir) within three days of the index date, COVID-19 vaccination status (unvaccinated, 1–2 doses, and ≥3 doses) and calendar week of index date. Patients were considered vaccinated if they had received the COVID-19 vaccine at least two weeks before the index date. The immunocompromised patients were those with diagnosed immunocompromising conditions (HIV, hematological malignancy, immune-mediated rheumatic disease, other hematological conditions, solid organ transplant, and bone marrow or stem cell transplant). Patients were also considered as immunocompromised if they had a history of receiving or had remaining days supply of a monoclonal antibody within the last 3 months, an oral immunosuppressive drug within the last 1 month, an oral glucocorticoid (equivalent to 20 mg/day of prednisone taken continuously) within the last 1 month, or if they had received an immunosuppressive infusion or injection within 3 months before the index date.[31]

## Statistical analysis
Following previous works, we implemented the target trial emulation through a three-step approach, including cloning, censoring, and weighting.[22,36] First, all eligible patients included for analysis were cloned at a 1:1 ratio to either the treatment group or the control group for each trial. Second, clones were censored if they deviated from the treatment protocol during the 5 days following SARS-CoV-2 diagnosis (i.e., the index date). This grace period aligns with antiviral treatment guidelines from the DH of Hong Kong and the US Food and Drug Administration, which recommend initiating nirmatrelvir/ritonavir or molnupiravir within 5 days of illness onset.[37,38] Clones in any of the antiviral treatment groups were censored if they did not receive the specific treatment within a 5-day grace period after the index date, while clones in the control group were censored when they received

either of the treatments during the 5 days. This procedure ensures that all study participants follow the assigned treatment strategy within a 5-day grace period after the index date, thereby addressing the immortal time bias.[39] Third, IPCW was then performed at each day during the grace period to address the selection bias resulting from the artificial censoring. Specifically, the probability of not being censored on each day was calculated by a multivariate logistic regression model adjusting for baseline covariates, conditional on remaining uncensored in the previous day. The inverse of the probability was further stabilized by the proportion of individuals remaining in the group, which was considered as the weight on that day. The cumulative multiplicative weights on the last day of the 5-day grace period were used for further analyses. Covariates with an absolute SMD ≥ 0.1 after the weighting procedure were considered imbalanced and were included in the corresponding models for doubly robust adjustment.[40]

The short- and long-term cumulative incidence of cardiovascular complications was estimated for each emulated trial, with adjustment for baseline covariates. Adjusted RD of cardiovascular outcomes was calculated for treatment and control arms for each trial, accounting for baseline covariates. Considering death as a potential competing event, a cause-specific hazard model with IPCW was used to estimate the HR between arms in each trial, with 95% CI computed using a nonparametric bootstrap of 1000 runs of resampling.[41,42] Subgroup analyses by age groups (≤65 years or >65 years), sex (female or male), and vaccination status (unvaccinated or ≥1 dose of vaccine) were conducted by repeating the analysis for each stratification. To examine the robustness of the primary results, we conducted sensitivity analyses by (1) defining the outcome assessment period as day 0–30 for short-term outcomes and day 31–365 for long-term outcomes; (2) restricting the analysis among patients without any of the outcome events diagnosed within 6 years before SARS-CoV-2 infection (in the main analyses, patients without a history the outcome events within 3 years before infection were included); and (3) applying the Fine-Gray model as an alternative method of cause-specific hazard model in the presence of competing risk.[43]

All statistical analyses were performed using **R** statistical software (version 4.2.2) (**R** Program for Statistical Computing). The "*dplyr*" (version 1.1.4), "*survival*" (version 3.7·0), and "*ggplot2*" (version 3.5.1) packages were used for data processing, analysis, and visualization. All statistical tests were two-tailed, and a *p*-value < 0.05 was considered a sign of statistical significance.

### Reporting summary
Further information on research design is available in the Nature Portfolio Reporting Summary linked to this article.

## Data availability
The Hong Kong Hospital Authority and Department of Health, the Government of the Hong Kong Special Administrative Region, are the data custodians, and data requests to these parties can be made via email (hacpaaedr@ha.org.hk) and website (https://www.dh.gov.hk/english/aboutus/aboutus_pps/aboutus_pps.html), respectively. The cases' surveillance data and medication records were extracted from electronic records in the system managed by the Hong Kong Hospital Authority. The vaccine history was extracted from the COVID-19 surveillance database provided by the Department of Health in Hong Kong. Restrictions apply to the availability of these data, which were used under an agreement for the purposes of scientific research. The authors do not have the right to transfer or release the data, in whole or in part, and in whatever form or media, or to any other parties or place outside of Hong Kong, and must fully comply with the duties under the law relating to the protection of personal data, including those under the Personal Data (Privacy) Ordinance and its principles in all aspects.

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

## Acknowledgements

We thank the Hospital Authority and the Department of Health, Hong Kong Government, for providing the data for this study. The Centre for Health Systems and Policy Research, funded by the Tung Foundatio,n is acknowledged for the support throughout the conduct of this study. This research was funded by Health and Medical Research Fund [EKY: grant numbers COVID190105, COVID19F03, INF-CUHK-1, COVID1903003], CUHK Direct Grant (KCC: 2025.059; YW: 4054934), National Natural Science Foundation of China (KCC: 72574190), RGC Collaborative Research Fund (CKPM: C6036-21GF), and RGC theme-based research schemes (DSCH: T11-705/21-N). Shi Zhao was funded by the National Natural Science Foundation of China (grant no.: 12401648), the Noncommunicable Chronic Diseases - National Science and Technology Major Project of China (grant no.: 2023ZD0519300), the Young Elite Scientists Sponsorship (YESS) Program by CAST (grant no.: 2024QNRC001), the Natural Science Foundation of Tianjin Municipal Science and Technology Commission (grant no.: 24JCQNJC00610), and the Tianjin Medical University start-up funding. The funders of the study had no role in study design, data collection, data analysis, data interpretation, writing of the manuscript, or the decision to submit for publication.

## Author contributions

Study design and conceptualization: Z.G., Y.W., K.M.J., C.B., S.Z., G.L., and K.C.C. Data collection and pre-processing: Z.G., Y.W., H.W., C.H.K.Y., T.Y.C., and E.K.Y. Data analysis and interpretation: Z.G., Y.W., G.L., C.L., S.Z., and K.C.C. Writing - Original Draft: Z.G., Y.W., G.L., K.M.J., H.W., C.T.H., S.Z., and K.C.C. Writing - Review and Editing: C.B., S.Z., C.H.K.Y., T.Y.C., C.L., KL, A.Y., C.K.P.M., D.S.C.H., and E.K.Y. Supervision: S.Z., E.K.Y. and K.C.C. have accessed and verified all the data. All authors critically reviewed the manuscript and gave final approval for publication.

## Competing interests

The authors declare no competing interests.

## Additional information

Zihao Guo[1,10], Yuchen Wei[1,10], Guozhang Lin[1], Katherine Min Jia[2], Christopher Boyer[3], Huwen Wang[4], Conglu Li[1],
Chi Tim Hung[1], Carrie Ho Kwan Yam[1], Tsz Yu Chow[1], Shi Zhao[5] ✉, Kehang Li[1], Aimin Yang[6], Chris Ka Pun Mok[1,7,8,9],
David SC Hui[6,8], Eng Kiong Yeoh[1] ✉ & Ka Chun Chong[1] ✉

[1]The Jockey Club School of Public Health and Primary Care, The Chinese University of Hong Kong, Hong Kong, China. [2]Center for Communicable Disease
Dynamics, Department of Epidemiology, Harvard T.H. Chan School of Public Health, Boston, MA, USA. [3]Cleveland Clinic Lerner College of Medicine, Case
Western Reserve University, Cleveland, OH, USA. [4]Duke-NUS Medical School, Singapore, Singapore. [5]School of Public Health, Tianjin Medical University,
Tianjin, China. [6]Department of Medicine & Therapeutics, Faculty of Medicine, The Chinese University of Hong Kong, Hong Kong, China. [7]Li Ka Shing Institute of
Health Sciences, Chinese University of Hong Kong, Hong Kong, China. [8]S.H. Ho Research Centre for Infectious Diseases, Chinese University of Hong Kong,
Hong Kong, China. [9]School of Biomedical Sciences, The Chinese University of Hong Kong, Hong Kong SAR, China. [10]These authors contributed equally: Zihao
Guo, Yuchen Wei. ✉e-mail: zhaoshi.cmsa@gmail.com; yeoh_ek@cuhk.edu.hk; marc@cuhk.edu.hk

