## [Peer Review file · Nature Communications]

Effectiveness of nirmatrelvir/ritonavir and molnupiravir in reducing the risk of short-term and long-term cardiovascular complications of COVID-19: a target trial emulation study

Corresponding Author: Dr Ka Chun Chong

Version 0:

Reviewer comments:

Reviewer #1

(Remarks to the Author)

The study used data from the Hong Kong Hospital Authority to evaluate the effectiveness of COVID-19 antivirals on the risk of post-COVID cardiovascular complications. The dataset is rich, and multiple analyses have been conducted. However, I have several concerns regarding the study design. However, I believe these issues could likely be addressed through revision.

1. While the authors correctly note that target trial emulation, if applied properly, can avoid immortal time bias. However, in this study treatment was assigned within 5 days of infection, whereas the true follow-up (outcome ascertainment) began 21 days after infection. Such misalignment has not been addressed although the study claim applying the target trial emulation. Please consider and address the situation of participants who died or experienced events within the first 21 days, given that the risk of these events is likely differ between the treatment and control? Application of IPCW for first 21 days or evaluate effect since index date could provide different causal estimand for the study.

2. The specification of the target trial and its emulation requires substantial modification to align with correct target trial emulation practices.

- a. Since this is not a three-arm study, participants are not assigned to one of three treatment strategies. The trial protocol should be conceptualized as the emulation of two separate two-arm trials with separated treatment assignment.
- b. With the application of the clone method, the study likely estimates a per-protocol effect rather than an intention-to-treat effect.
- c. Please specify the causal contrast the study aims to estimate, particularly given that outcome ascertainment begins 21 days after the index date.
- d. I recommend that the authors carefully review the target trial emulation framework to ensure the study is conducted appropriately. Please also consider updating the reporting of the target trial and its emulation following the recently published guideline (DOI: 10.1136/bmj-2025-087179).

3. In both figure and results section, please estimate and report the adjusted risk difference, rather than the unadjusted risk difference.

4. Page 7, lines 254–255, the study reports enrolling participants until October 10, 2023, with data available through October 31, 2023. Under these conditions, it seems unexpected that the follow-up period could range from 358 to 365 days. Please consider provide a clearer description of the actual follow-up duration.

5. Table 1 shows a study population with median age above 70. Please clarify whether this age distribution reflects the general population hospitalized for COVID-19.

Also in Table 1, the prevalence of comorbidities such as dementia, cancer, Charlson Comorbidity Index and more appears to be low given the age distribution (more than 50% with age above 70). Could the authors provide some explanations?

Reviewer #2

(Remarks to the Author)

This is a well-conducted and original study addressing a clinically important question: the effectiveness of nirmatrelvir/ritonavir and molnupiravir in reducing the risk of post-COVID cardiovascular complications. The study is methodologically sound, based on high-quality data, and employs appropriate statistical techniques. The conclusions are clearly presented and well-supported. The manuscript is clearly written, with relevant references and strong overall coherence.

Comments for Improvement:

- It is not clear whether a study protocol for the target trial emulation was pre-registered. Please clarify this point.
- The manuscript does not mention whether actual randomized controlled trials (RCTs) on this topic exist. The introduction seems to focus solely on observational studies. Have the authors reviewed existing RCTs? Please write an additional section on this issue.
- It remains unclear whether the target trial emulation was designed to replicate an existing RCT or to model a hypothetical one. This distinction should be clarified.

Version 1:

Reviewer comments:

Reviewer #1

(Remarks to the Author)

I thank the authors for the improvement and revision of the manuscript. The current version shows substantial improvement in alignment with the target trial emulation framework.

I have no further comments.

REVIEWER COMMENTS

Reviewer #1 (Remarks to the Author):

The study used data from the Hong Kong Hospital Authority to evaluate the effectiveness of COVID-19 antivirals on the risk of post-COVID cardiovascular complications. The dataset is rich, and multiple analyses have been conducted. However, I have several concerns regarding the study design. However, I believe these issues could likely be addressed through revision.

Response: We thank the reviewer for the positive comments.

1. While the authors correctly note that target trial emulation, if applied properly, can avoid immortal time bias. However, in this study treatment was assigned within 5 days of infection, whereas the true follow-up (outcome ascertainment) began 21 days after infection. Such misalignment has not been addressed although the study claim applying the target trial emulation.

Please consider and address the situation of participants who died or experienced events within the first 21 days, given that the risk of these events is likely differ between the treatment and control? Application of IPCW for first 21 days or evaluate effect since index date could provide different causal estimands for the study.

Response: Thank you for your constructive comments. We agree with the reviewer that it is important to ensure the alignment of the eligibility determination, treatment assignment, and follow-up initiation in a target trial emulation study design. In this revision, we thus redefined the outcome variables as post-COVID complications which were assessed from the index date (positive RT-PCR date of SARS-CoV-2 infection).

To address potential differential risk arises from the acute phase and post-acute phase of infection to and to differentiate the post-acute COVID-19 conditions from acute conditions, we categorized outcomes into short-term outcomes and long-term outcomes, where the former was evaluated through day 21 post-infection and the latter was evaluated from day 22-365 among patients who were alive at day 22, to address the informative censoring caused by death within the first 21 days as mentioned by the reviewer. This analysis strategy was commonly adopted a previous target trial emulation studies (<https://doi.org/10.7326/M22-3565>; <https://doi.org/10.7326/M23-1394>). Therefore, the causal estimand of the study changed to the cumulative incidence of antiviral agents (nirmatrelvir/ritonavir or molnupiravir) versus no-treatment on day 0-21 and day 22-365 risk of cardiovascular complications in hospitalized patients with COVID-19.

We reanalysed the data accordingly and we found that while both nirmatrelvir/ritonavir arm or molnupiravir arm were associated with lower risk for post-COVID complications in short-

term compared to the corresponding control arm, only nirmatrelvir/ritonavir-treated patients demonstrated a risk reduction for post-COVID outcomes in long-term.

Revised results section:

Lines 364-431:

“Short-term outcomes

Effect of nirmatrelvir/ritonavir

The median follow-up period for patients in nirmatrelvir/ritonavir trial was 21 days for short-term outcomes. The 21-day adjusted cumulative incidence of cardiovascular death, MACE, and composite cardiovascular complications were shown in Figure 2. There were reductions in the risk of cardiovascular mortality (adjusted RD (95% CI) = -0.97% (-1.24 to -0.71); adjusted HR (95%CI) = 0.45 (0.28 to 0.72)), composite cardiovascular complications (adjusted RD = -3.02% (-3.64 to -2.41); adjusted HR = 0.55 (0.46 to 0.65)), MACE (adjusted RD = -2.13% (-2.58 to -1.67); adjusted HR = 0.47 (0.39 to 0.57)), cerebrovascular disorders (adjusted RD = -0.91% (-1.26 to -0.57); adjusted HR = 0.43 (0.31 to 0.60)), dysrhythmia (adjusted RD = -0.67% (-1.03 to -0.31); adjusted HR = 0.68 (0.56 to 0.83)), ischemic heart disease (adjusted RD = -1.14% (-1.45 to -0.83); adjusted HR = 0.38 (0.28 to 0.53)), and other cardiac disorders (adjusted RD = -0.87% (-1.12 to -0.63); adjusted HR = 0.51 (0.32 to 0.81)). The 21-day risk of inflammatory heart disease and thrombotic disorders was similar between arms (Figure 3).

Effect of molnupiravir

The median follow-up period for patients in molnupiravir trial was 21 days for short-term outcomes. Molnupiravir was associated with a lower risk of cardiovascular mortality (adjusted RD = -1.03% (-1.37 to -0.69); adjusted HR = 0.66 (0.53 to 0.82)), composite cardiovascular complications (adjusted RD = -1.36% (-2.03 to -0.70); adjusted HR = 0.80 (0.72 to 0.90)), MACE (adjusted RD = -1.02% (-1.52 to -0.53); adjusted HR = 0.75 (0.65 to 0.87)), cerebrovascular disorders (adjusted RD = -0.74% (-1.06 to -0.43); adjusted HR = 0.57 (0.44 to 0.76)), and thrombotic disorders (adjusted RD = -0.16% (-0.29 to -0.02); adjusted HR = 0.54 (0.31 to 0.96)). The risk of dysrhythmia, ischemic heart disease, other cardiac disorders, and inflammatory heart disease was similar between arms (Figure 2 and Figure 3).

Long-term outcomes

Effect of nirmatrelvir/ritonavir

The median follow-up period for patients in nirmatrelvir/ritonavir trial was 365 days for long-term outcomes. In general, the 22- to 365-day risk of cardiovascular outcomes were lower in nirmatrelvir/ritonavir arm than in control arm. Compared with no-treatment group, the use of nirmatrelvir/ritonavir was associated with a

significantly lower risk of cardiovascular mortality (adjusted RD = -0.81% (-1.19 to -0.43); adjusted HR = 0.61 (0.47 to 0.79)), composite cardiovascular complications (adjusted RD = -1.50% (-2.15 to -0.85); adjusted HR = 0.73 (0.63 to 0.86)), MACE (adjusted RD = -1.23% (-1.80 to -0.67); adjusted HR = 0.70 (0.56 to 0.87)), cerebrovascular disorders (adjusted RD = -0.53% (-0.89 to -0.16)); adjusted HR = 0.66 (0.51 to 0.86)), dysrhythmia (adjusted RD = -0.73% (-1.10 to -0.35); adjusted HR = 0.63 (0.52 to 0.77)), ischemic heart disease (adjusted RD = -0.21% (-0.49 to 0.08); adjusted HR = 0.75 (0.60 to 0.95)), and other cardiac disorders (adjusted RD = -0.84% (-1.28 to -0.39); adjusted HR = 0.70 (0.51 to 0.95)). The statistical significances of inflammatory heart disease and thrombotic disorders reached the marginal level of 0.05 (Figure 4 and Figure 5).

Effect of molnupiravir

The median follow-up period for patients in molnupiravir trial was 365 days for long-term outcomes. Compared to nirmatrelvir/ritonavir, the benefits of molnupiravir on cardiovascular outcomes were less apparent (Figure 4 and Figure 5). Among all the cardiovascular outcomes, molnupiravir was only marginally significantly associated with a decreased risk of cardiovascular mortality (adjusted RD = -0.37% (-0.78 to 0.03); adjusted HR = 0.81 (0.68 to 0.96)), compared to the control group (Figure 5)."

2. The specification of the target trial and its emulation requires substantial modification to align with correct target trial emulation practices.
 - a. Since this is not a three-arm study, participants are not assigned to one of three treatment strategies. The trial protocol should be conceptualized as the emulation of two separate two-arm trials with separated treatment assignment.

Response: Thank you for the comment. We have revised the specifications of target trials and their emulation following the recent published TARGET (The Transparent Reporting of Observational Studies Emulating a Target Trial) guideline (DOI: 10.1136/bmj-2025-087179). The revised trial specification was detailed in Table S1, where we conceptualized the trial protocol as two separate two-arm trials with separated treatment assignment.

Lines 190-194:

"The study followed the TARGET (The Transparent Reporting of Observational Studies Emulating a Target Trial) guideline [18] and the STROBE (Strengthening the Reporting of Observational Studies in Epidemiology) reporting guideline. Ethics approval was obtained from the Joint CUHK-NTEC Clinical Research Ethics Committee (No. 2023.006). No study protocol was registered for this study."

b. With the application of the clone method, the study likely estimates a per-protocol effect rather than an intention-to-treat effect.

Response: Thank you for pointing out the issue. We have revised the causal contrast as per-protocol effect, as shown in Table S1.

c. Please specify the causal contrast the study aims to estimate, particularly given that outcome ascertainment begins 21 days after the index date.

Response: Thank you. We have revised the causal contrast as per-protocol effect, as shown in Table S1. the causal estimand of the study changed to the cumulative incidence of antiviral agents (nirmatrelvir/ritonavir or molnupiravir) versus no-treatment on day 0-21 and day 22-365 risk of cardiovascular complications in hospitalized patients with COVID-19.

d. I recommend that the authors carefully review the target trial emulation framework to ensure the study is conducted appropriately. Please also consider updating the reporting of the target trial and its emulation following the recently published guideline (DOI: 10.1136/bmj-2025-087179).

Response: Thank you for your suggestion. We have revised the manuscript extensively following the TARGET (The Transparent Reporting of Observational Studies Emulating a Target Trial) guideline (DOI: 10.1136/bmj-2025-087179), with a TARGET checklist submitted alongside the revised manuscript.

Lines 190-194:

“The study followed the TARGET (The Transparent Reporting of Observational Studies Emulating a Target Trial) guideline [18] and the STROBE (Strengthening the Reporting of Observational Studies in Epidemiology) reporting guideline. Ethics approval was obtained from the Joint CUHK-NTEC Clinical Research Ethics Committee (No. 2023.006). No study protocol was registered for this study.”

3. In both figure and results section, please estimate and report the adjusted risk difference, rather than the unadjusted risk difference.

Response: Thank you. We have replaced the crude risk difference with estimated adjusted risk difference in all figures and in the result section.

4. Page 7, lines 254–255, the study reports enrolling participants until October 10, 2023, with data available through October 31, 2023. Under these conditions, it seems unexpected that the follow-up period could range from 358 to 365 days. Please consider provide a clearer description of the actual follow-up duration.

Response: Thank you. We have rephrase the description regarding the follow-up period for patients in lines 366, 381, 409, and 425 of the revised manuscript:

“The median follow-up period for patients in nirmatrelvir/ritonavir trial was 21 days for short-term outcomes.”

“The median follow-up period for patients in molnupiravir trial was 21 days for short-term outcomes.”

“The median follow-up period for patients in nirmatrelvir/ritonavir trial was 365 days for long-term outcomes.”

“The median follow-up period for patients in molnupiravir trial was 365 days for long-term outcomes.”

5. Table 1 shows a study population with median age above 70. Please clarify whether this age distribution reflects the general population hospitalized for COVID-19. Also in Table 1, the prevalence of comorbidities such as dementia, cancer, Charlson Comorbidity Index and more appears to be low given the age distribution (more than 50% with age above 70). Could the authors provide some explanations?

Response: Thank you for the comments. The median age of all hospitalized patients with COVID-19 in Hong Kong during the observational period of our study is 77 years old [IQR = 66–87], obtained from the source database retrieved from Hong Kong Hospital Authority (HA). As stated in the main text, the HA served over 7.3 million residents and handling more than 90% of all hospitalizations in Hong Kong. Therefore, the age distribution for patients included in our study likely reflect the general population hospitalized for COVID-19 in Hong Kong.

As noted in our manuscript, a key inclusion criterion of study participants was the absence of pre-existing cardiovascular diseases. Given that cardiovascular diseases is strongly associated with and often a precursor to a multitude of other serious conditions—including chronic kidney disease, diabetes, and certain cancers—it is possible that a cohort free of CVD would also demonstrate a lower overall burden of these related comorbidities. In addition, the distribution of comorbidities were similar to our previous study investigating the effectiveness of nirmatrelvir–ritonavir against post-acute sequelae of COVID-19 among

COVID-19 hospitalizations who are immunocompetent (DOI: 10.1016/S2665-9913(24)00224-8).

To this end, we have mentioned these point in the limitation section.

Lines 799-817:

“Fourthly, although the median age of our study participants was consistent with the that of the general population hospitalized for COVID-19 in Hong Kong (77 years [IQR = 66–87] obtained from the original database), future study conducted in a younger population is warranted to validate our findings. In addition, as the absence of cardiovascular diseases was one of the inclusion criteria, the prevalence of comorbidities of study participants would be lower than general population, given that cardiovascular diseases were associated with conditions including chronic kidney disease, diabetes, and certain cancers.”

Reviewer #2 (Remarks to the Author):

This is a well-conducted and original study addressing a clinically important question: the effectiveness of nirmatrelvir/ritonavir and molnupiravir in reducing the risk of post-COVID cardiovascular complications. The study is methodologically sound, based on high-quality data, and employs appropriate statistical techniques. The conclusions are clearly presented and well-supported. The manuscript is clearly written, with relevant references and strong overall coherence.

Response: We thank the reviewer for the positive comments. Based on the first comment from reviewer #1, we have revised our definition of outcome measures, which included both short-term (from 0-21 days after index date) and long-term (from 22-365 days after index date) cardiovascular complications (originally, only the long-term outcomes were analysed). Therefore, we extensively revised the text while the main conclusion remained the similar as the original manuscript.

Comments for Improvement:

- It is not clear whether a study protocol for the target trial emulation was pre-registered. Please clarify this point.

Response: Thank you. As this is an observational study for trial emulation, there was no pre-registered study protocol. However, the ethics approval has been obtained, and we have followed the recent published TARGET (The Transparent Reporting of Observational Studies Emulating a Target Trial) guideline to do the study reporting.

Lines 190-194:

“The study followed the TARGET (The Transparent Reporting of Observational Studies Emulating a Target Trial) guideline [18] and the STROBE (Strengthening the Reporting of Observational Studies in Epidemiology) reporting guideline. Ethics approval was obtained from the Joint CUHK-NTEC Clinical Research Ethics Committee (No. 2023.006). No study protocol was registered for this study.”

- The manuscript does not mention whether actual randomized controlled trials (RCTs) on this topic exist. The introduction seems to focus solely on observational studies. Have the authors reviewed existing RCTs? Please write an additional section on this issue.

Response: Thank you for the comment. We have done a literature review of published RCTs relevant to the topic of the study, and we found no RCT has been conducted to evaluate the efficacy of antivirals against post-infection cardiovascular complications in COVID-19 hospitalizations. We have added an additional section in the introduction on this point.

Lines 111-120:

“Several randomized control trials have been initiated to evaluate the nirmatrelvir/ritonavir as a potential treatment for post-acute sequelae of COVID-19 [12-15]. The RECOVER-Vital [12], PAX LC trial [13], and STOP-PASC trial [14] focused on patients with long COVID while the CanTreatCOVID involved patients with acute infection [15]. However, these trials only focused on non-specific conditions. To date, there is no randomized control trial specifically investigating the efficacy of antiviral agents on cardiovascular complications in COVID-19 hospitalizations. Here, we used a target trial emulation study design to evaluate the effectiveness of nirmatrelvir/ritonavir and molnupiravir on acute and post-acute COVID-19 cardiovascular complications among the hospitalized population.”

- It remains unclear whether the target trial emulation was designed to replicate an existing RCT or to model a hypothetical one. This distinction should be clarified.

Response: Thank you. As there is no existing RCT for addressing the research question of this study, we conducted a target trial emulation study to model a hypothetical one. We have mentioned this in the revised introduction section.

Lines 111-120:

“Several randomized control trials have been initiated to evaluate the nirmatrelvir/ritonavir as a potential treatment for post-acute sequelae of COVID-19 [12-15]. The RECOVER-Vital [12], PAX LC trial [13], and STOP-PASC trial [14] focused on patients with long COVID while the CanTreatCOVID involved patients with acute infection [15]. However, these trials only focused on non-specific conditions. To date, there is no randomized control trial specifically investigating the efficacy of antiviral agents on cardiovascular complications in COVID-19 hospitalizations. Here, we used a target trial emulation study design to evaluate the effectiveness of nirmatrelvir/ritonavir and molnupiravir on acute and post-acute COVID-19 cardiovascular complications among the hospitalized population.”

REVIEWERS' COMMENTS

Reviewer #1 (Remarks to the Author):

I thank the authors for the improvement and revision of the manuscript. The current version shows substantial improvement in alignment with the target trial emulation framework.

I have no further comments.

Response: Thank you for your positive comment.